# The potential role of cuproptosis-related genes for therapy and immunoregulation in pan-cancer

Jianpeng Zhou[1], Chuanlei Wang[1], Yao Zhi[1], Jia Li[2]*

**1** Department of Hepatobiliary and Pancreatic Surgery I, General Surgery Center, The First Hospital of Jilin University, Changchun, Jilin Province, China, **2** Department of Hematology, The First Hospital of Jilin University, Changchun, Jilin Province, China

* lijia0810@jlu.edu.cn

## Abstract

The primary drawbacks of current cancer therapies are lower selectivity for cancer cells, more side effects, and obscure resistance mechanisms. Novel approaches to overcome these drawbacks comprise the utilization of ionophores and metalliferous chelators to change the concentration of trace metal elements in cancer cells. As the concept of cuproptosis emerged, it might be a novel strategy to enhance the curative effects for resistant cancer cells potentially. FDX1, LIAS, LIPT1, DLD, DLAT, PDHA1, PDHB, and SLC31A1 are the major regulators of cuproptosis. However, the expression landscape and clinical roles of these regulators remain to be addressed. This study explored the expression pattern and clinical role of these cuproptosis-related genes in pan-cancer by evaluating the association of tumor mutation burden (TMB), immune-related scores, cells in tumor microenvironment, and drug sensibility. The results displayed that the expressions of cuproptosis-related genes were significantly different in various cancer types, all cuproptosis-related gene upregulates significantly in LAML, ALL, PAAD, GBM, GBMLGG, LGG, and all significantly downregulated in cancers KIPP, WT, KIPAN, KIRC. Furthermore, the higher the level of cuproptosis-related genes expressed, the higher the survival in patients suffering from KIRC, and KIPAN increased. In addition, the expression of cuproptosis-related genes was negatively associated with immune-related scores, while SLC31A1 had a positive association with StromalScore, ImmuneScore, and EstimateScore in LAML. Importantly, the expression of cuproptosis-related genes was positively correlated with common lymphoid progenitor (CLP) cells and Th2 cells, but negatively associated with NKT cells or Th1 cells. These findings suggest that cuproptosis-related genes are dysregulated across cancer types, hold prognostic value, and may be involved in modulating the tumor immune microenvironment.

**Data availability statement:** The data used in this study are publicly available and were obtained from the UCSC Xena platform (https://xenabrowser.net/datapages/), which provides access to standardized datasets from The Cancer Genome Atlas (TCGA). The specific datasets include: • RNA-seq gene expression data (HTSeq - FPKM) from the GDC TCGA Hub • Somatic mutation data from MuTect2 Variant Aggregation and Masking • Curated clinical phenotype data from the Pan-Cancer Atlas Hub • Immune subtype data from the Pan-Cancer Atlas Hub All datasets are freely available and accessible without restriction or registration. The UCSC Xena project is managed by the University of California, Santa Cruz, and governed under public data-sharing policies.

**Funding:** The author(s) received no specific funding for this work.

**Competing interests:** The authors have declared that no competing interests exist.

## Introduction

Copper(Cu) is an essential cofactor in most organisms, which sustains at lower levels through active homeostasis mechanisms to hold back intracellular dissociative copper amassing regulated by concentration gradients. Thus, an overload of intracellular copper concentration leads to cell death. Recently, Todd R. Golub's team demonstrated that the mechanisms of copper toxicity are different from other forms of cell death, such as apoptosis, ferroptosis, pyroptosis, and necrosis, and termed this previously uncharacterized cell death as cuproptosis [1], which is induced by cooper-dependent lipoylated proteins oligomerization and Fe-S cluster proteins instability. The study also suggests that cuproptosis might be regarded as a helpful and promising therapy against cancer.

Elevated copper concentrations have been observed in various tumor tissues compared to their normal counterparts. Accumulation of copper has been associated with tumor proliferation, angiogenesis, and metastasis, suggesting that dysregulated copper metabolism plays a role in cancer development and progression. Furthermore, Cu levels have been found significantly altered in both serums and tumor tissues of patients with tumors such as oral, bladder, cervical, breast, ovarian, thyroid, pancreatic, prostate, gastric, and lung tumors [2–11]. Therefore, targeting copper homeostasis may represent a novel therapeutic strategy in oncology.

Recent advances have suggested that modulating copper-dependent cell death could be leveraged to treat cancer. Both copper ionophores and chelators have been proposed as potential agents to selectively induce cytotoxicity in cancer cells while sparing normal tissue. However, the functional roles of cuproptosis in different tumor contexts—particularly its involvement in shaping the tumor immune microenvironment—remain largely unclear.

Several genes have been identified as key regulators of the cuproptosis pathway, including *FDX1, LIAS, LIPT1, DLD, DLAT, PDHA1, PDHB,* and *SLC31A1*. While these genes have been mechanistically linked to copper-induced cell death, their pan-cancer expression profiles and clinical relevance are not fully characterized. In addition, recent reviews have raised questions regarding the consistency and significance of cuproptosis in cancer immunity and microenvironment regulation [12], highlighting the need for comprehensive studies to clarify these mechanisms.

In this study, we aimed to systematically investigate the expression landscape of cuproptosis-related genes across 34 cancer types using large-scale genomic datasets. We analyzed their association with tumor mutation burden (TMB), microsatellite instability (MSI), neoantigen load (NEO), tumor-infiltrating immune cell populations, and drug sensitivity. Furthermore, we explored their potential prognostic value and impact on the tumor microenvironment. This work provides novel insights into the functional relevance of cuproptosis-related genes and may offer new avenues for personalized cancer therapy.

## Materials and methods

### Cuproptosis-related genes collection

According to Todd R. Golub's study, seven positive regulatory genes FDX1, LIAS, LIPT1, DLD, DLAT, PDHA1, PDHB, and a Cu transporter encoding gene SLC31A1 were collected for the pan-cancer analysis [1].

### Data resources

The FANTOM5(60 tissue types) [13], Human Protein Atlas(HPA, 40 tissue types) [14], and GTEx database(37 tissue types) [15] were used to download the gene expression data of human tissues. The UCSC(https://xenabrowser.net/datapages/) was the data source for standardized universal omics data, which included gene expression RNAseq from HTSeq-FPKM GDC Hub, somatic mutation from MuTect2 Variant Aggregation and Masking, clinical phenotype from Curated clinical data by Pan-Cancer Atlas Hub, immune phenotype from immune subtype by Pan-Cancer Atlas Hub. The maftools R package was used to calculate the distribution of tumor mutation burden (TMB) according to somatic mutation data. The MSI score was obtained from the previous study [16]. The NEO score was obtained from the previous study [17]. To analyze cuproptosis-related genes expression, 34 cancer types were utilized, including adrenocortical carcinoma (ACC), acute lymphoblastic leukemia(ALL),bladder urothelial carcinoma (BLCA), breast invasive carcinoma (BRCA), cervical squamous cell carcinoma and endocervical adenocarcinoma (CESC), cholangiocarcinoma(CHOL), colon adeno-carcinoma (COAD), colon adenocarcinoma/rectum adenocarcinoma esophageal carcinoma(COADREAD), esophageal carcinoma (ESCA), glioblastoma multiforme (GBM), glioma(GBMLGG), head and neck squamous carcinoma (HNSC), kidney chromophobe (KICH), kidney renal clear cell carcinoma (KIRC), kidney renal papillary cell carcinoma (KIRP), Pan-kidney cohort(KIPAN),acute myeloid leukemia (LAML), brain lower grade glioma (LGG), liver hepatocellular carcinoma (LIHC), lung adenocarcinoma (LUAD), lung squamous cell carcinoma (LUSC), ovarian serous cystadenocarcinoma (OV), pancreatic adenocarcinoma (PAAD), pheochromocytoma and paraganglioma (PCPG), prostate adenocarcinoma (PRAD), rectum adenocarcinoma (READ), skin cutaneous melanoma (SKCM), stomach adenocarcinoma (STAD), Stomach and Esophageal carcinoma(STES),testicular germ cell tumors (TGCT), thyroid carcinoma (THCA), uterine corpus endometrial carcinoma (UCEC), uterine carcinosarcoma (UCS) and High-Risk Wilms Tumor(WT). All abbreviations are shown in S1 Table. All statistical R packages are shown in S2 Table.

### Evaluation of differential cuproptosis-related genes expression between tumor and normal tissues

The cuproptosis-related gene expression level was extracted from the RNAseq dataset. The differential expression genes(DEGs) between tumor and normal tissues across 34 cancer types were analyzed by ggpubr R package, with statistical significance (adjusted p-value <0.05). The significance of gene expression alterations was identified by the Wilcox method. The p-value was adjusted with the Benjamini-Hochberg multiple testing correction. The heatmap of cupro ptosis-related genes was plotted by pheatmap R packages.

### Correlation analysis of cuproptosis-related genes expression with TMB, MSI and NEO

The corrplot R package was applied to evaluate the correlation between the cuproptosis-related gene expression and TMB, MSI, or NEO with the Spearman method ($p < 0.05$). The correlation radar plot was plotted by the ggradar R package, and the ggpubr R package was applied to plot boxplots.

### Survival analysis of cuproptosis-related gene expressions

The tumor samples were divided into high- and low-expression groups according to the median value of cuproptosis-related gene expression across 34 cancer types. The survminer R package was applied for overall survival analysis by the Kaplan-Meier method, for which the log-rank test was used, with a statistical significance of $p < 0.05$. The survival R

package was used to perform Cox regression analysis for cuproptosis-related genes. The hazard ratio was calculated for the Cox proportional hazard regression models. Further, the differential expression of cuproptosis-related genes in different pathologic stages (including stages I, II, III, and IV) were analyzed across 34 cancer types.

### Correlation analysis of cuproptosis-related genes expression with immune microenvironment across 34 cancer types

The presence of infiltrating stromal and immune cell scores was predicted by the ESTIMATE algorithm in the estimate R package that forecasted stromal and immune cells in tumor tissues with gene expression data. Based on ssGSEA analysis, the estimate algorithm generated ImmuneScore, StromalScore, and EstimateScore. These three scores represent the corresponding ratio of immune cells infiltrating in tumor tissues, stromal cells present in tumor tissues, and the sum of both, respectively. Furthermore, the higher score represents the larger ratio of the corresponding component in the tumor microenvironment. Correlation analysis of the cuproptosis-related gene expressions with ImmuneScore, StromalScore, or EstimateScore was performed by the Corrplot R package with the method of Spearman (p < 0.05).

### The proportion of cells in microenvironment across 34 cancer types based on xCell method

The xCell algorithm was employed to identify the proportion of cells in the microenvironment across 34 cancer types. The xCell R package was applied to discriminate 64 human cell phenotypes in the microenvironment based on the gene expression profile. The correlation analysis of the cuproptosis-related gene expressions with different cells was performed by the corrplot R package with the Spearman method (p < 0.05), including activated dendritic cell (aDC), adipocytes, astrocytes, B cells, basophils, CD4 + memory T cells, CD4 + naïve T cells, CD4 + T cells, CD4 + central memory T cells(CD4 + Tcm), CD4 + effector memory T cells(CD4 + Tem), CD8 + naïve T cells, CD8 + T cells, CD8 + T cells, CD8 + central memory T cells(CD8 + Tcm), CD8 + effector memory T cells(CD8 + Tem), conventional dendritic cells(cDC), chondrocytes, class switched memory B cells, common lymphoid progenitor cells(CLP), common myeloid progenitor(CMP), dendritic cells(DC), endothelial cells, eosinophils, epithelial cells, erythrocytes, fibroblasts, granulocyte-macrophage progenitor(GMP), hepatocytes, hematopoietic stem cells(HSC), immature dendritic cells(iDC), Keratinocytes, lymphatic(ly) endothelial cells, macrophages, M1 macrophages, M2 macrophages, mast cells, megakaryocytes, melanocytes, memory B cells, megakaryocyte-erythroid progenitor(MEP), mesangial cells, monocytes, multipotent progenitors(MPP), mesenchymal stem cells(MSC), microvascular(mv) endothelial cells, myocytes, naïve B cells, neurons, neutrophils, NK cells, natural killer T cells(NKT), osteoblasts, plasma dendritic cells(pDC), pericytes, plasma cells, platelets, preadipocytes, pro B cells, sebocytes, skeletal muscle cells, smooth muscle cells, gamma delta T cells, T helper 1 cells(Th1 cells), T helper 2 cells(Th2 cells), T regulator cells(Treg cells).

### Drug sensitivity analysis

CellMiner database (https://discover.nci.nih.gov/cellminer/) was used to explore transcript and drug patterns in the NCI-60 cell line set. The NCI-60 cell line panel was an anticancer drug efficacy screen by the Developmental Therapeutics Program (DTP) of the US National Cancer Institute (NCI). Thousands of compounds have been applied to the NCI-60. The sample of gene expression and drug sensitivity data were downloaded from the CellMiner database and then filtered drug sensitivity data after clinical laboratory verification and FDA standard certification. Next, the Spearman correlation test was performed for cuproptosis-related gene expression data combined with drug sensitivity data.

## Results

### The expression of cuproptosis-related genes in human tissues and cancers

The expression of cupro ptosis-related genes was analyzed in human normal tissues and cancer samples from FANTOM5, HPA, and GTEx databases. All database data showed concordant results that cuproptosis-related genes were highly

expressed in heart muscle tissue. and data from FANTOM5 and the HPA database showed that cuproptosis-related genes were highly expressed in the kidney (Fig 1A–1C). In addition, cuproptosis-related is widely expressed in various cancer types (Fig 1D). Further, cuproptosis-related genes showed extensive alterations in tumor samples compared with adjacent samples across most of cancer types, such as ACC, ALL, BLCA, BRCA, CHOL, COAD, COADREAD, ESCA, GBM, GBMLGG, HNSC, KICH, KIPAN, KIRC, KIRP, LAML, LGG, LIHC, LUAD, LUSC, OV, PAAD, PRAD, READ, STES, STAD, TGCT, THCA, UCS and WT (Fig 1E, S1A–S1H Fig). Most of the cuproptosis-related genes show significant upregulation compared with adjacent non-tumor tissues across different cancer types. As can be seen, all cuproptosis-related gene upregulates significantly in LAML, ALL, PAAD, GBM, GBMLGG, and LGG and all significantly downregulated in cancers KIPP, WT, KIPAN, and KIRC (Fig 1E). These results might indicate that different cuproptosis-related genes play different roles in various cancer types.

## The potential prognostic value of cuproptosis-related genes expression in different cancer types

To evaluate the prognostic potential of cuproptosis-related genes across 34 cancer types, the Kaplan-Meier curve analysis based on TCGA, GTEx, and TARGET database was performed. Most of the genes showed significant correlations with the overall survival of patients. KIRC and KIPAN showed a lot of significant results. The higher the level of cuproptosis-related genes expressed, the higher the survival rate in patients suffering from KIRC, and KIPAN increased (Fig 2A–2P). However, for other cancers, different cuproptosis-related genes played different roles. To be more specific, the higher the level of FDX1 expressed, the poorer survival in ALL, GBMLGG, LAML, and LGG (S2A–S2D Fig). The increasing expression of LIAS has a significant correlation with poor survival in KICH, LAML, and THCA, while better survival in GBMLGG (S2E–S2H Fig). The increasing expression of LIPT1 has a significant correlation with pool survival in GBMLGG, LIHC, and LAML, while better survival in BLCA, READ, and SKCM (S2I–S2N Fig). The increasing expression of DLAT has a significant correlation with poor survival rate in BLCA, BRCA, GBMLGG, LGG, LIHC, and PAAD, but better survival in COAD, COADREAD, and READ (S2O–S2X Fig). The increasing expression of DLD was highly correlated to poor survival rates in BRCA, GBMLGG, LGG, and LUAD, but better survival in COADREAD (S3A–S3G Fig). The increasing expression of PDHA1 was highly correlated to poor survival in ESCA, LAML, LUAD, PRAD, and SKCM (S3H–S3M Fig). The increasing expression of PDHB was highly correlated to poor survival in KICH, and LAML but better survival in KIRP (S3N–S3P Fig). The increasing expression of SLC31A1 was highly correlated to poor survival in ACC, BLCA, BRCA, GBMLGG, LAML, and LGG (S3Q–S3V Fig).

Furtherly, to obtain hazard ratio (HR) across 34 cancer types for cuproptosis-related genes, the univariate Cox regression analysis was performed (S4 Fig). These results further confirmed Kaplan-Meier curves of overall survival analysis, from which the indication that the same gene might be a risky factor (HR > 1) or protective factor (HR < 1) in different tumors. Taking SLC31A1 as an exaple, SLC31A1 played risky role in BLCA (HR = 1.28, p = 0.02), BRCA (HR = 1.31, p = 0.01), GBMLGG (HR = 3.25, p = 2.00E-21) and LGG (HR = 2.38, p = 6.70E-07) but played protective role in KIRC (HR = 0.68, p = 3.70E-06) and KIPAN (HR = 0.76, p = 1.20E-04).

In addition, correlation analyses of pathologic stages (stages I, II, III, and IV) and cuproptosis-related genes were performed across 34 cancer types. The expression levels of a cuproprosis-related gene are significantly different among different stages in ACC, CESC, GBMLGG, KIRP, STAD, STES, and UCS. Detailly, the level of LIAS has a significant difference among different stages in UCS, and ACC, higher expression levels indicated increasing pathologic stage (Fig 3A). Lower expression of LIPT1 might suggest an increasing pathologic stage (Fig 3B). For the expression of DLAT among different stages, it showed a neutral difference in KIRP (Fig 3C). The lower expression of PDHA1 was, the lower stage of CESC, while no trend was shown in STAD and STES (Fig 3D). The level of SLC31A1 also has differences among different stages in cancers GBMLGG, and KIRP (Fig 3E).

## The mutation frequency of cuproptosis-related genes

The detailed mutation status of cuproptosis-related genes was displayed by the waterfall map from high to low percentage, LIAS, PDHA1, DLAT, DLD, LIPT1, SLC31A1, PDHB, FDX1 (Fig 4). There were many types of mutations to be found,

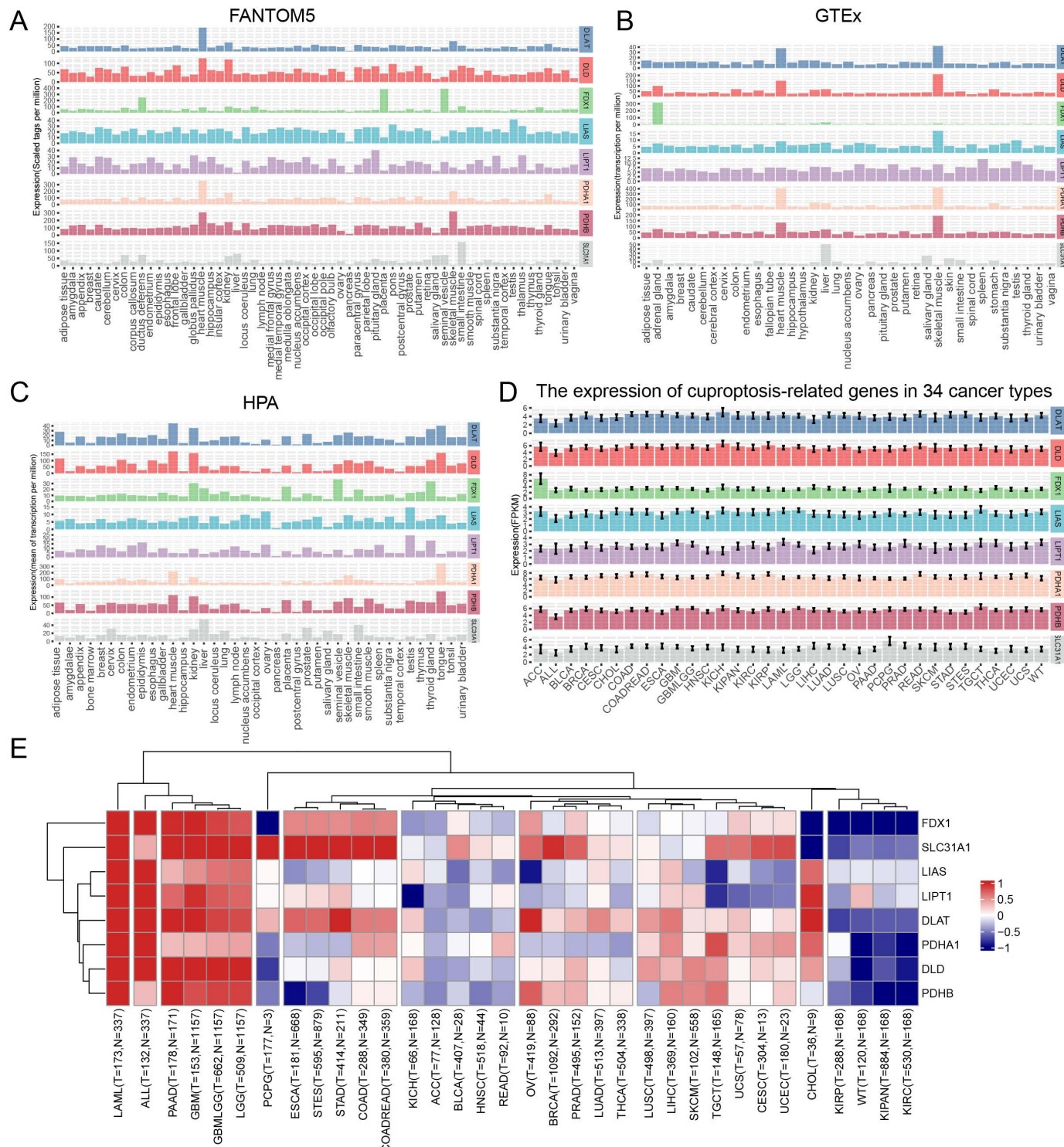

**Fig 1. The level of cuproptosis-related gene expressions in human normal tissues and tumor samples.** The RNA-seq data of cuproptosis-related genes in human tissues from the **(A)** FANTOM5 database(60 tissues), **(B)** GTEx database(37 tissues), and **(C)** HPA database(40 tissues). **(D)** The RNA-seq data of cuproptosis-related genes in various cancers from UCSC. **(E)** The heatmap of widespread alterations of cuproptosis-related genes across 34 cancer types compared with adjacent or normal samples.

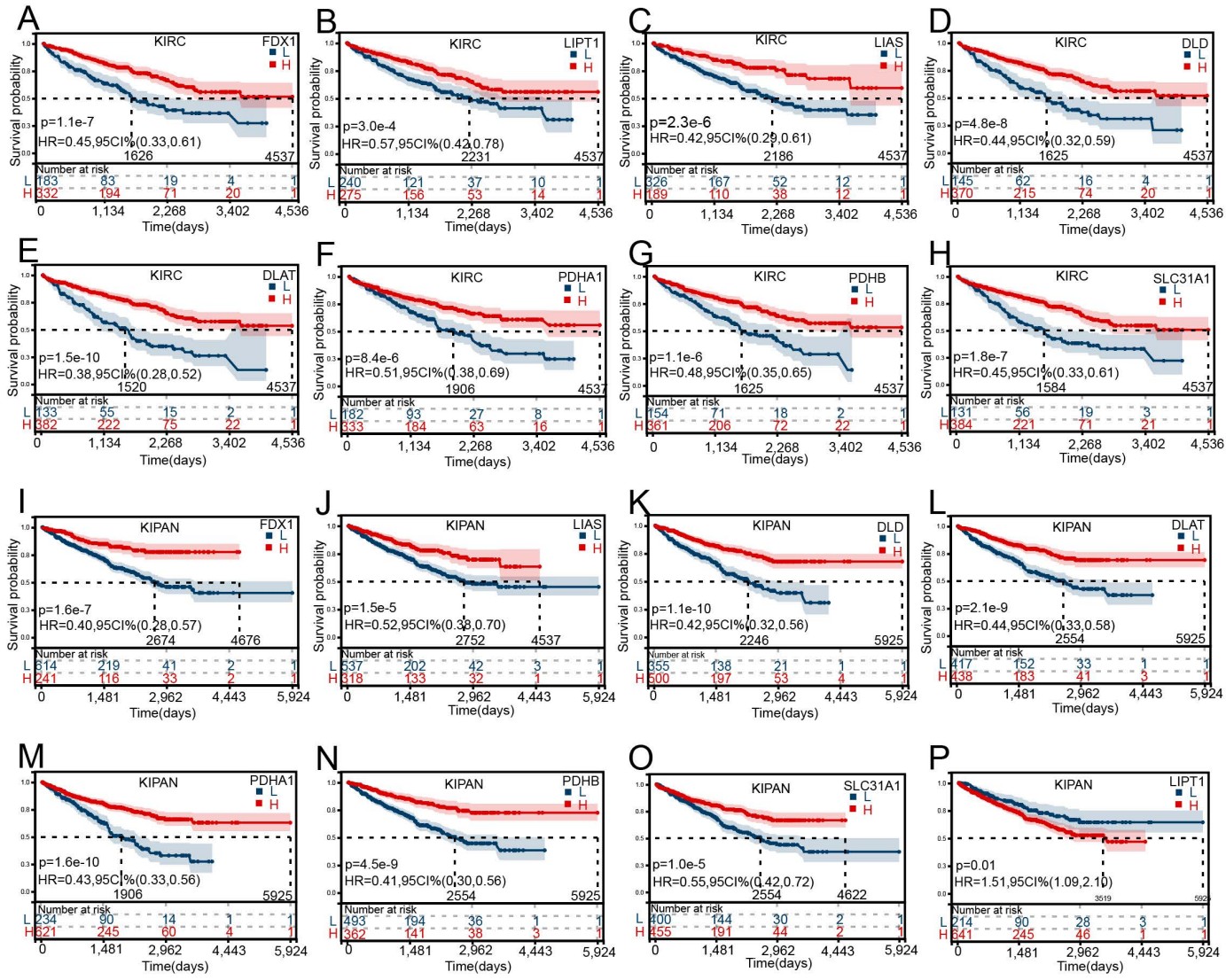

**Fig 2. The overall survival analysis for cuproptosis-related gene expression in KIRC and KIPAN by the Kaplan-Meier curves.** The overall survival analysis for the expression(the median expression as a cut-off) of **(A)** FDX1, **(B)**LIPT1, **(C)**LIAS, **(D)**DLD, **(E)** DLAT, **(F)** PDHA1, **(G)** PDHB, **(H)** SLC31A1 in KIRC. The overall survival analysis for the expression(the median expression as a cut-off) of **(I)** FDX1, **(J)** LIAS, **(K)** DLD, **(L)** DLAT, **(M)** PDHA1, **(N)** PDHB, **(O)** SLC31A1, **(P)**LIPT1 in KIPAN.

including frameshift variant, intron variant, missense variant, 3′ prime UTR variant, 5′ prime UTR variant, synonymous variant, downstream gene variant, splice region variant, splice donor variant, stop gained, inframe deletion, protein-altering variant, splice acceptor variant, start lost, stop lost, stop retained variant, and upstream gene variant.

## Correlation analyses of the cuproptosis-related gene expressions and TMB or MSI or NEO

The role of TMB (Tumor mutation burden) in tumors has been more and more important in recent years and it has been regarded as a novel biomarker in tumors. Correlation analysis of the expression of the cupro ptosis-related gene with TMB index was performed for various cancer types (Fig 2A). Cuproptosis-related genes have a significant correlation with TMB scores in most cancers. However, different cancers correlated differentially with cuproptosis-related genes. For example,

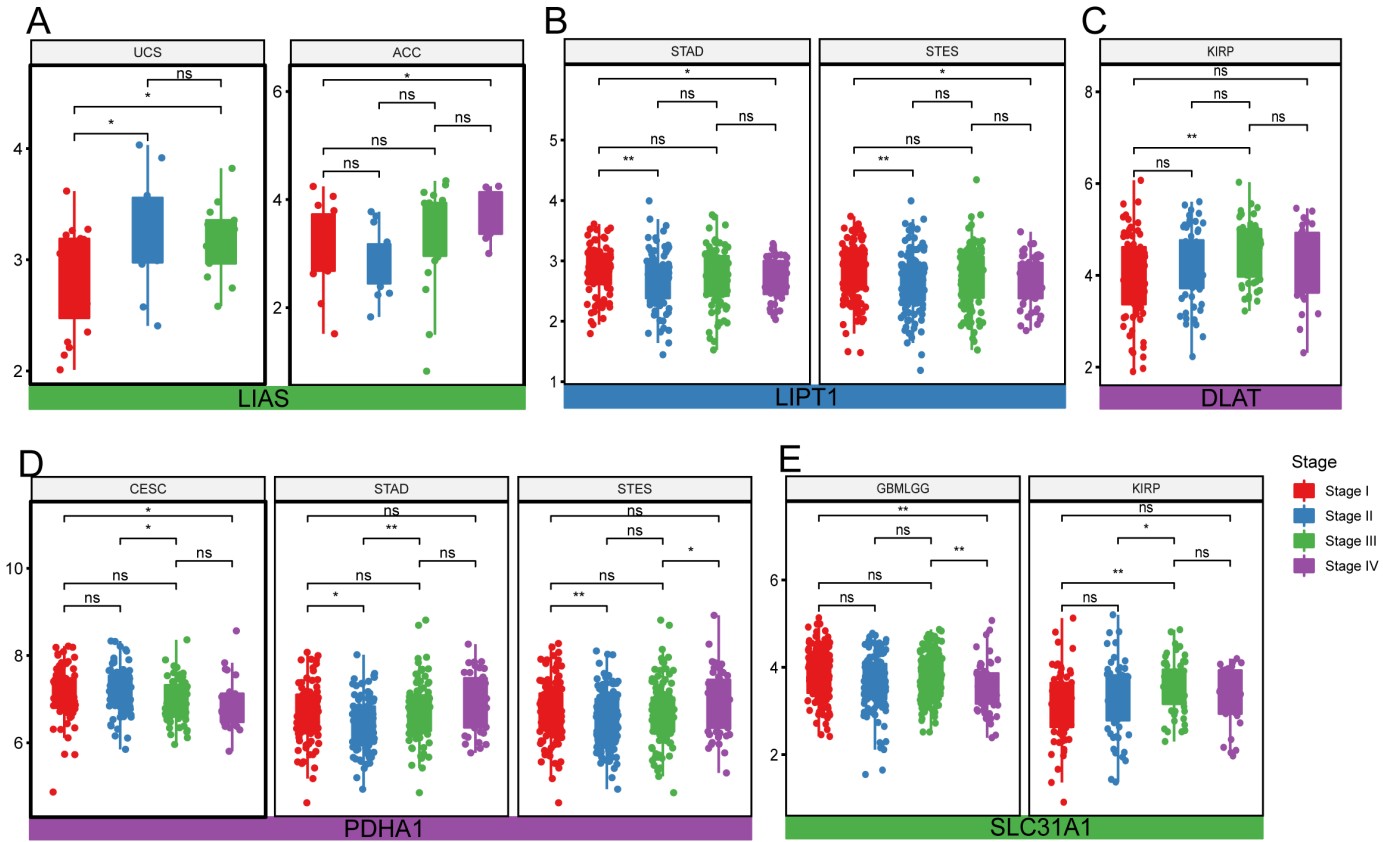

**Fig 3. The difference of cuproptosis-related gene expressions among different stages in various cancer types. (A)** LIAS expressed among different stages UCS, ACC. **(B)** LIPT1 is expressed among different stages in STAD, and STES. **(C)** DLAT is expressed among different stages in KIRP. **(D)** PDHA1 is expressed among different stages in CESC, STAD, and STES. **(E)** SLC31A1 expressed among different stages in GBMLGG, KIRP.

SLC31A1 exhibited a positive correlation with TMB score in ACC, BRCA, COAD, COADREAD, and STAD but a negative correlation with KIPAN, and PRAD (Fig 5A, S5AS5H Fig). MSI (Microsatellite instability) has also attracted more and more attention in recent years. The correlation analyses of the cuproptosis-related gene expressions and MSI index were also performed in various tumor types (Fig 5B, S6AS6H Fig). Almost all cuproptosis-related genes have a significant correlation with MSI index in different tumor types.

Neoantigens (NEO) are tumor-specific antigens derived from non-synonymous mutations and have become a very attractive target for tumor immunotherapy, They are highly expressed in tumor cells with strong immunogenicity and tumor heterogeneity. Therefore, to perform the correlation analysis of the Cuproptosis-related gene expressions and NEO was useful for clinical. The results showed that strong positive correlations exist between NEO and Cuproptosis-related genes in ACC and UCES, and negative correlations exist in COAD, COADREAD, LUAD, PCPG, and THCA. These results further confirmed that Cuproptosis-related genes may affect antitumor immunity using regulating TMB, MSI, and NEO (Fig 5C, S7AS7H Fig).

## Cuproptosis-related gene expressions is associated with cells infiltrating in TME across various cancer types

The immune microenvironment plays a significant role in cancer progression. Whether cuproptosis-related gene expressions correlated with the immune microenvironment was explored in various cancer types by analyzing the ImmuneScore,

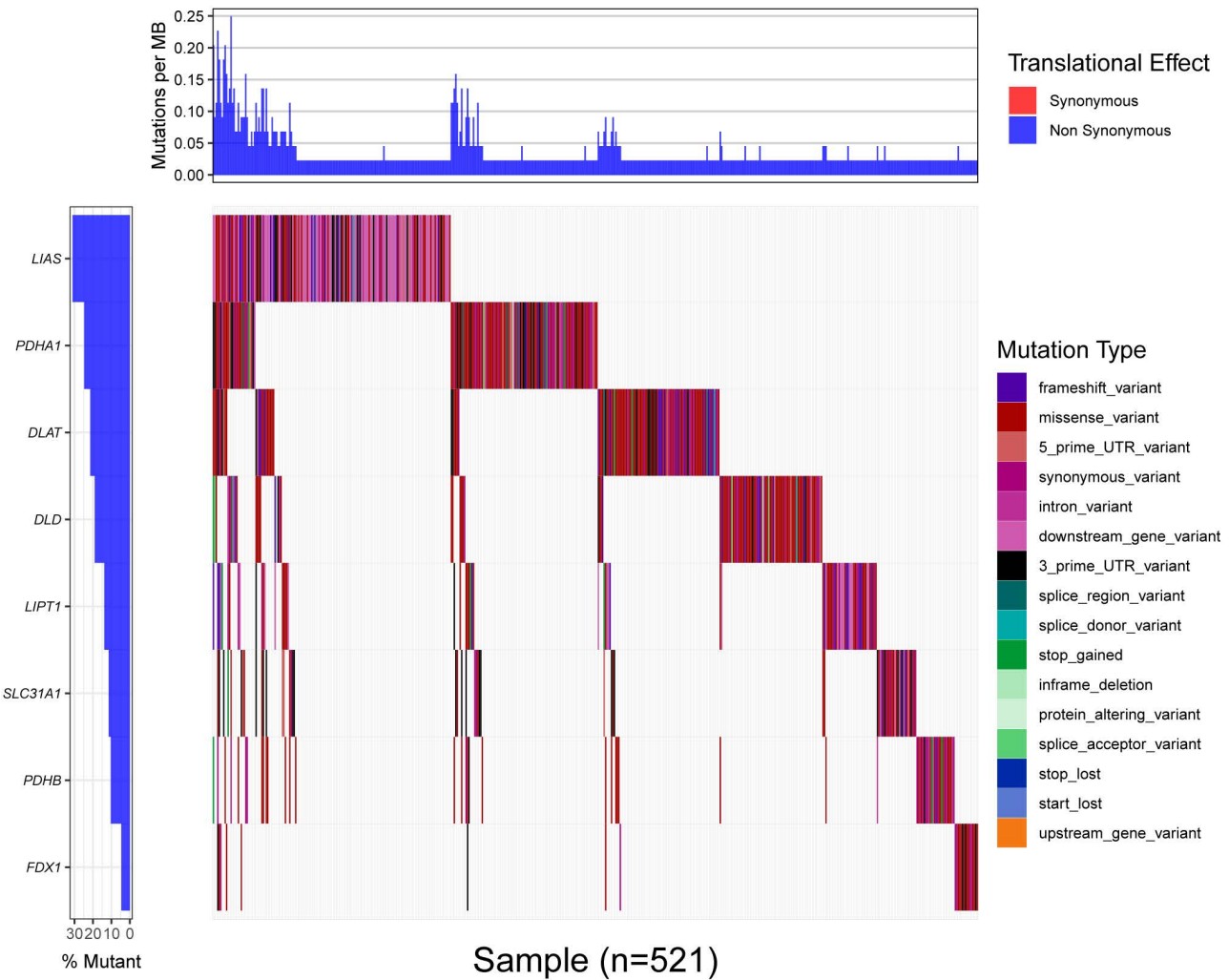

**Fig 4. The mutation frequency of Cuproptosis-related genes.**

StromalScore, EstimateScore, and cells in the tumor microenvironment (TME). The study showed that there was a significant association of the same cuproptosis-related gene with immune-related scores in various cancers; especially, most cuproptosis-related genes had negative correlations with immune-related scores in different cancers (Fig 6A), while SLC31A1 had positive association with StromalScore, ImmuneScore and EstimateScore in LAML (Fig 6B–6D).In addition, there were significant correlations between the expressions of cuproptosis-related genes and cells in TME (S8AS8H Fig), which suggested that cuproptosis might have a capacity to regulate the cells in TME.

Furthermore, the study proved that the level of cuproptosis-related gene expressions is positively associated with CLP cells or Th2 cells, but negatively associated with NKT cells or Th1 cells (Fig 7).

## Association of cuproptosis-related gene expressions with drug sensibility

To explore the correlation between cuproptosis-related genes and drug sensibility, a correlation analysis was performed. The results showed that the drugs included MI-503, BY-87–2243, Crizotinib, RX-3117, tic10, and AT13387 have significantly positive association with cuproptosis-related genes (Fig 8A–8B) (correlation coefficient > 0.43).

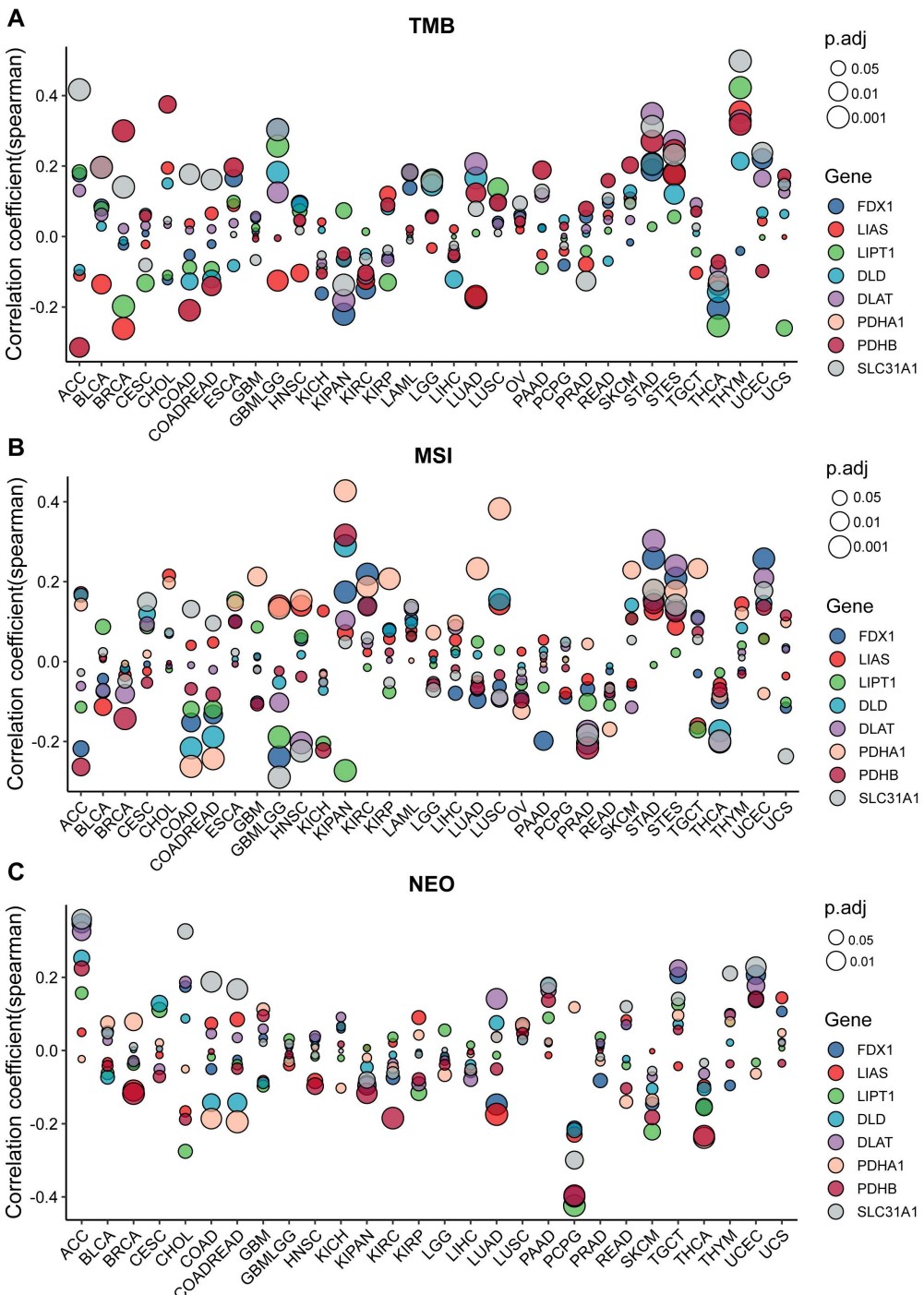

**Fig 5. The correlation analyses of the cuproptosis-related gene expressions and TMB, MSI, and NEO. (A)** The association of cuproptosis-related gene expressions with TMB. **(B)** The association of cuproptosis-related gene expressions with MSI. **(C)** The association of cuproptosis-related gene expressions with NEO.

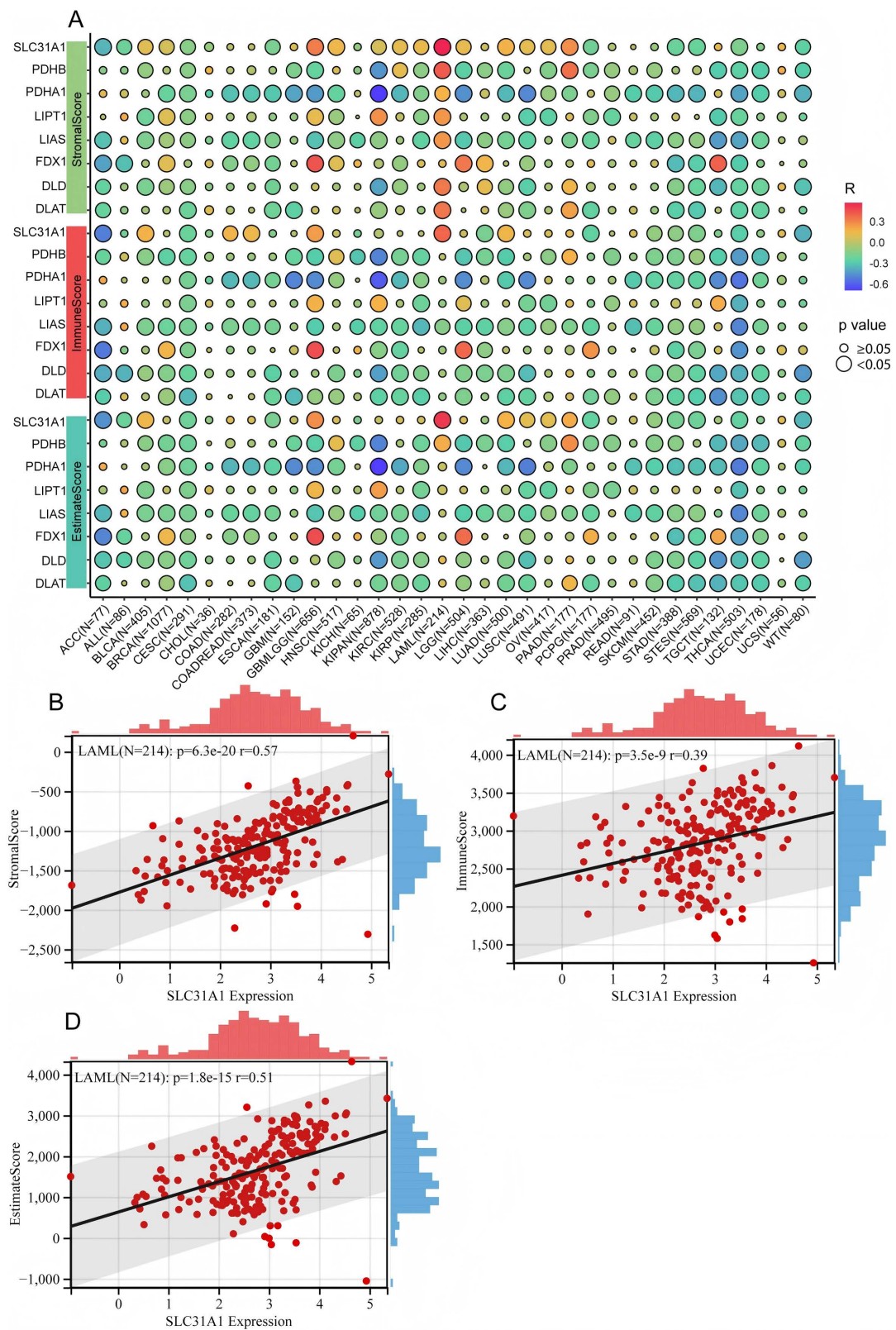

**Fig 6. Cuproptosis-related gene expressions associated with ImmuneScore, EstimateScore, and StromalScore in different cancers. (A)** Cuproptosis-related gene expressions associated with immune-related scores in different cancers. The correlations between SLC31A1 expression and **(B)** StromalScore, **(C)** ImmuneScore, **(D)** EstimateScore in LAML.

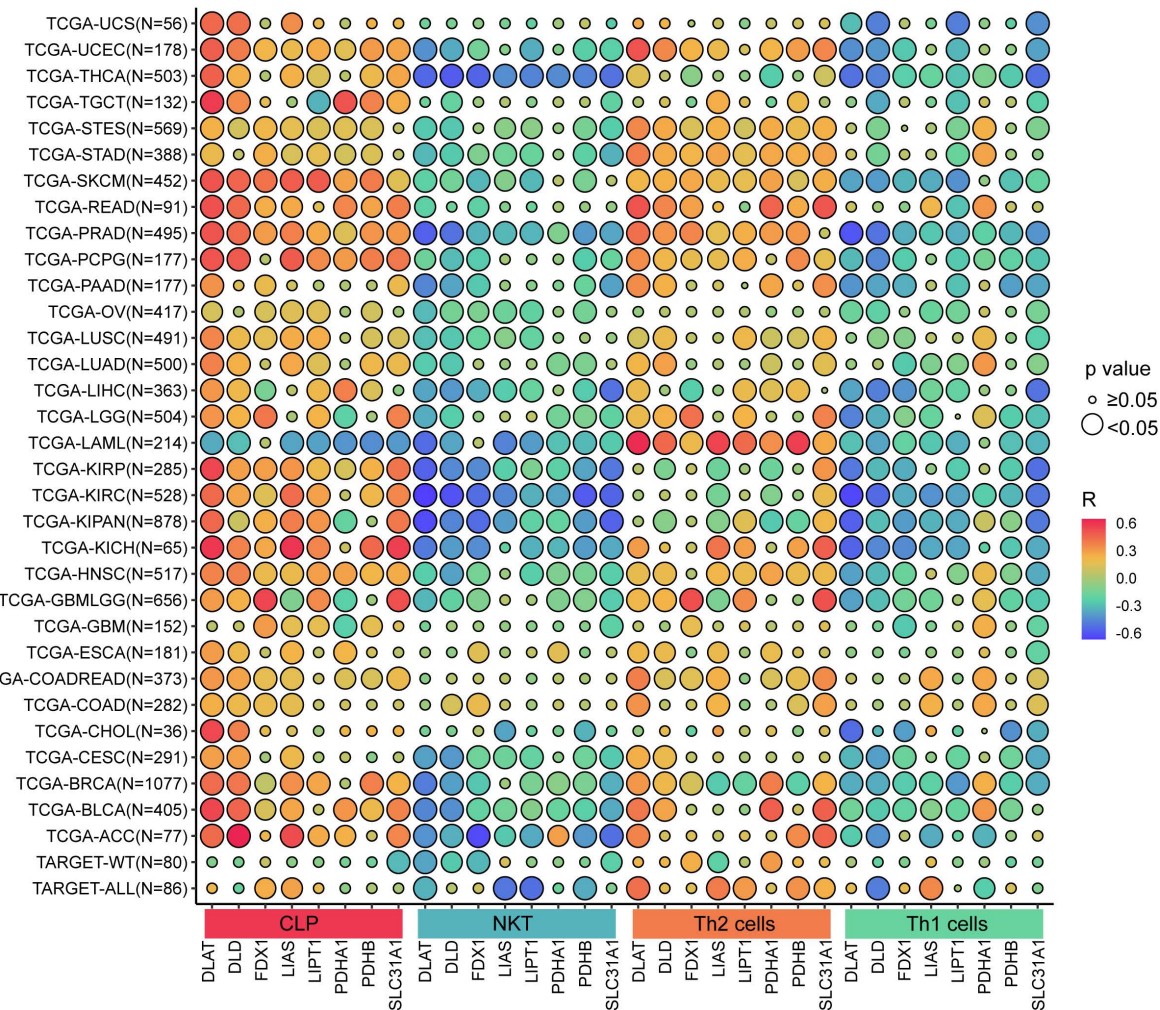

**Fig 7. The associations between cuproptosis-related gene expressions and cells in tumor microenvironment.**

## Discussion

The emergence of cuproptosis as a novel form of regulated cell death has opened new avenues for cancer therapy, particularly through the use of copper ionophores and copper-chelating agents. However, a key question remains: which cancer types are most susceptible to cuproptosis-inducing therapies, and through what mechanisms do cuproptosis-related genes influence tumor biology?

In this pan-cancer analysis, we systematically profiled the expression and prognostic value of eight well-characterized cuproptosis-related genes—*FDX1, LIAS, LIPT1, DLD, DLAT, PDHA1, PDHB*, and *SLC31A1*—across 34 cancer types. These genes were initially identified as core regulators of copper-induced cell death [1], but their functional roles in cancer immunity and therapeutic sensitivity have remained largely underexplored.

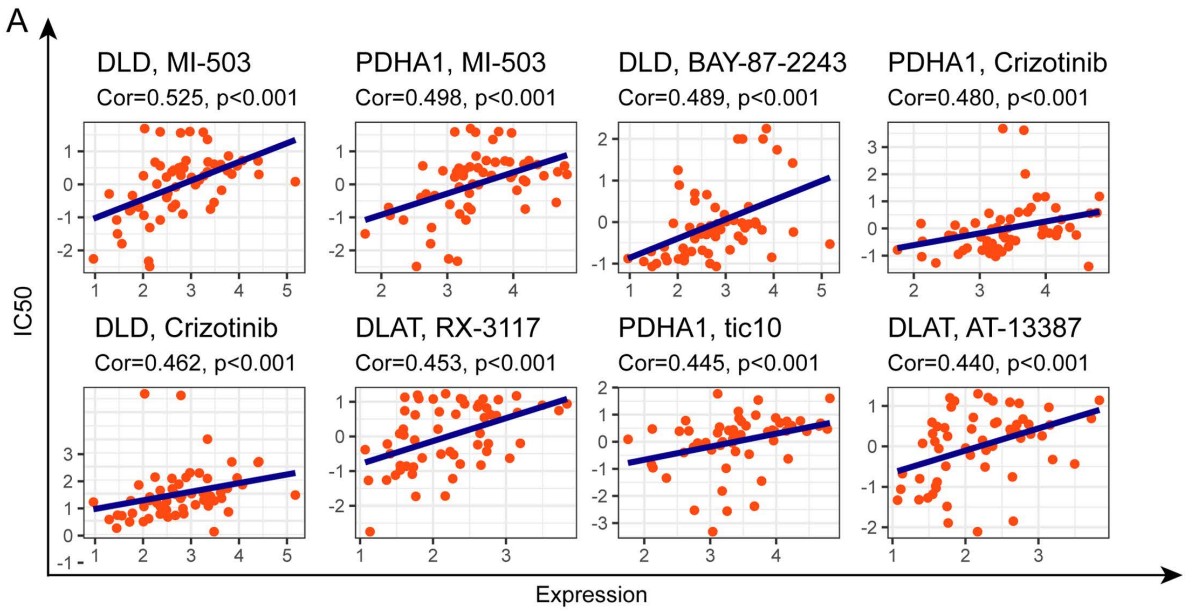

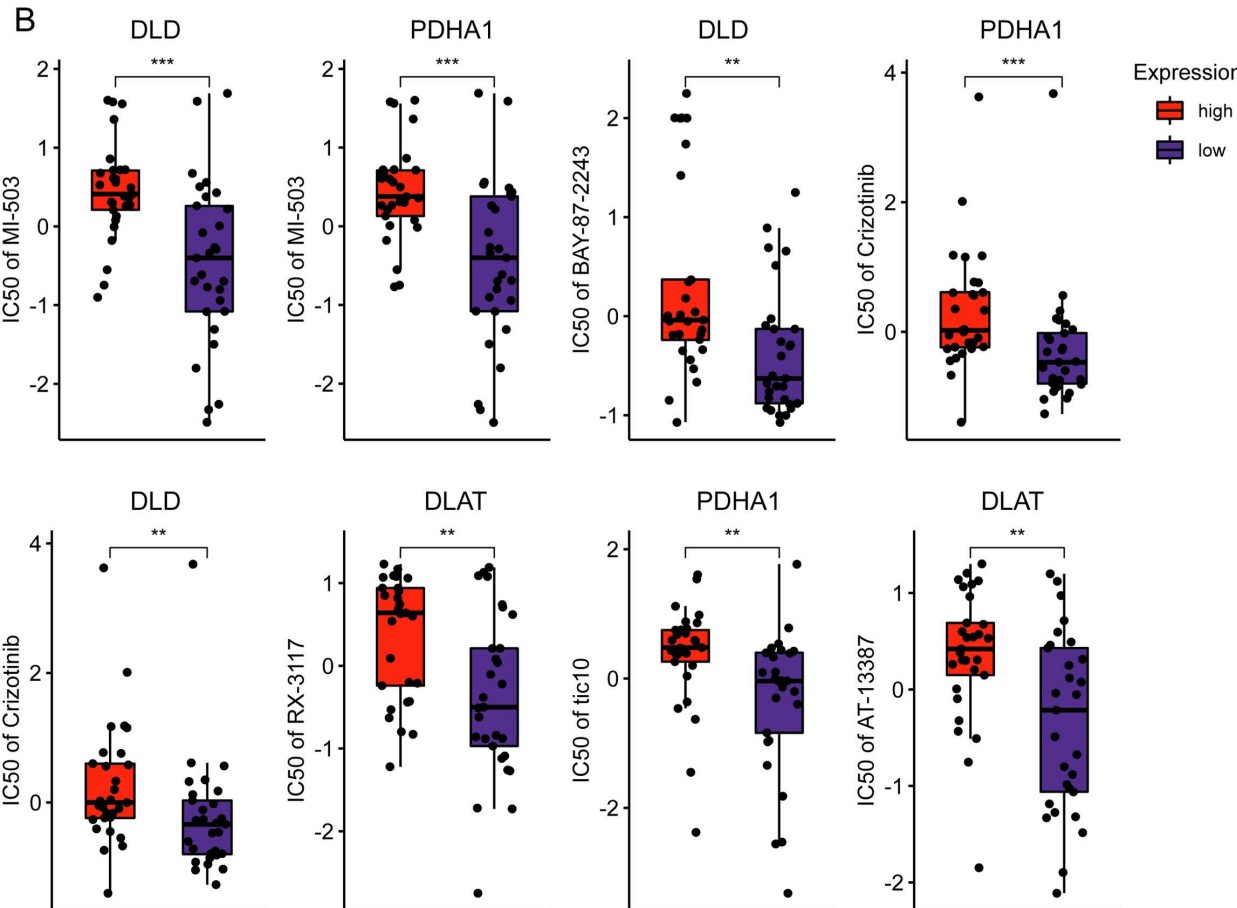

**Fig 8. Cuproptosis-related genes associated with drug sensibility. (A)** Cuproptosis-related genes correlated with drug sensibility. **(B)** The difference between the level of cuproptosis-related gene expression and the IC50 of various drugs.

We found widespread dysregulation of these genes in tumor samples compared to normal tissues. Notably, their upregulation in LAML, ALL, PAAD, GBM, GBMLGG, and LGG was associated with poor overall survival, whereas in KIRC and KIPAN, higher expression levels predicted better prognosis. These dual roles suggest that cuproptosis-related genes may have cancer-type-specific functions, acting as either oncogenic drivers or tumor suppressors depending on the cellular context.

Among all genes analyzed, *FDX1* and *SLC31A1* emerged as particularly relevant. *FDX1*, a mitochondrial reductase, has been reported to initiate cuproptosis by reducing Cu(II) to the more toxic Cu(I), thereby driving lipoylated protein aggregation. Its overexpression was linked to poor outcomes in several malignancies such as LAML and LGG. These findings imply that *FDX1* may be a promising therapeutic target, especially in tumors with mitochondrial metabolic dependencies. Moreover, its correlation with immune cell exclusion in the tumor microenvironment (TME) highlights a potential role in immunomodulation.

*SLC31A1*, the primary copper transporter encoded by the CTR1 gene, was found to be significantly associated with immune-related scores, particularly in LAML. Its expression positively correlated with StromalScore, ImmuneScore, and EstimateScore, suggesting a role in shaping the immune landscape of tumors. Interestingly, *SLC31A1* showed both risk and protective prognostic associations in different cancers, indicating complex, tissue-specific functions that merit further investigation. Its potential utility as a biomarker for copper uptake and sensitivity to copper-based therapeutics also deserves attention.

Our study further demonstrated that cuproptosis-related gene expression was significantly correlated with immune cell populations in the TME. Specifically, we found positive associations with common lymphoid progenitor (CLP) cells and Th2 cells, and negative associations with NKT and Th1 cells. This pattern suggests that cuproptosis may be involved in polarizing immune responses toward an anti-inflammatory phenotype, potentially contributing to immune escape in certain cancers.

Despite the insights gained, our study has limitations. First, although we included large-scale pan-cancer data, experimental validation in specific cancer types is needed to confirm the functional relevance of these findings. Second, while prior studies have explored cuproptosis-related genes across cancers, our work emphasizes immune correlation and identifies distinctive gene-immune interactions—particularly involving *FDX1* and *SLC31A1*—which were underreported in previous literature. Finally, additional genes involved in copper metabolism (e.g., *ATP7A, ATP7B, DBT, DLST*) should be explored in future studies to develop a more comprehensive model.

In summary, we identified several cuproptosis-related genes with strong potential as prognostic biomarkers and immunomodulatory targets in cancer. Focusing on genes such as *FDX1* and *SLC31A1* may guide the development of precision therapies that exploit copper-dependent vulnerabilities and reshape the immune landscape of tumors.

## Conclusions

Through comprehensive pan-cancer analyses, we demonstrated that cuproptosis-related genes exhibit widespread dysregulation across various cancer types and are significantly associated with patient prognosis, tumor immune microenvironment, and drug sensitivity. Our findings highlight the context-dependent roles of these genes—particularly *FDX1* and *SLC31A1*—in cancer progression and immune regulation.

The observed associations with tumor mutation burden (TMB), microsatellite instability (MSI), and neoantigen load (NEO) suggest that cuproptosis-related genes may influence tumor immunogenicity and serve as potential biomarkers for immunotherapy responsiveness. Moreover, their correlation with immune cell infiltration patterns—including CLP cells, Th2 cells, NKT cells, and Th1 cells—indicates a potential role in shaping immune responses within the tumor microenvironment.

Importantly, several cuproptosis-related genes showed significant associations with sensitivity to clinically relevant anti-cancer drugs, supporting their utility in guiding precision medicine strategies. Genes such as *FDX1* and *SLC31A1* may be

particularly promising candidates for targeted therapies aimed at modulating copper metabolism or enhancing copper-induced cell death in tumors.

In conclusion, this study provides a valuable resource for understanding the clinical and biological significance of cuproptosis-related genes in cancer. Future work integrating mechanistic experiments and functional validations in specific tumor types will be essential to confirm these findings and advance the development of cuproptosis-based therapeutic interventions.

## Supporting information

**S1 Fig. Boxplot of cuproptosis-related genes differential expression between cancer and adjacent normal tissues. (A)** FDX1 differential expression between cancer and adjacent normal tissue. **(B)** LIAS differential expression between cancer and adjacent normal tissue. **(C)** LIPT1 differential expression between cancer and adjacent normal tissue. **(D)** DLD differential expression between cancer and adjacent normal tissue. **(E)** DLAT differential expression between cancer and adjacent normal tissue. **(F)** PDHB differential expression between cancer and adjacent normal tissue. **(G)** PDHA1 differential expression between cancer and adjacent normal tissue. **(H)** SLC31A1 differential expression between cancer and adjacent normal tissue.
(TIF)

**S2 Fig. The Kaplan-Meier curves of overall survival in various cancers for cuproptosis-related gene expression.** The Kaplan-Meier curves of overall survival for the expression(the median expression as a cut-off) of FDX1 in **(A)** ALL, **(B)** GBMLGG, **(C)** LAML, and **(D)** LGG. The Kaplan-Meier curves of overall survival for the expression(the median expression as a cut-off) of LIAS in **(E)** KICH, **(F)** LAML, **(G)** THCA, **(H)** GBMLGG. The Kaplan-Meier curves of overall survival for the expression(the median expression as a cut-off) of LIPT1 in **(I)** BLCA, **(J)** GBMLGG, **(K)** LAML, **(L)** LIHC, **(M)**READ, **(N)** SKCM. The Kaplan-Meier curves of overall survival for the expression(the median expression as a cut-off) of DLAT in **(O)**BLCA, **(P)** BRCA, **(Q)** COAD, **(R)** COADREAD, **(S)** GBMLGG, **(T)** LAML, **(U)** LGG, **(V)** LIHC, **(W)** PAAD, **(X)** READ.
(TIF)

**S3 Fig. The Kaplan-Meier curves of overall survival in various cancers for cuproptosis-related gene expression.** The Kaplan-Meier curves of overall survival for the expression(the median expression as a cut-off) of DLD in **(A)** BRCA, **(B)** COADREAD, **(C)** GBMLGG, **(D)** KIRP, **(E)**LAML, **(F)** LGG, **(G)** LUAD. The Kaplan-Meier curves of overall survival for the expression(the median expression as a cut-off) of PDHA1 in **(H)** ESCA, **(I)** KIRP, **(J)** LAML, **(K)** LUAD, **(L)** PRAD, **(M)** SKCM. The Kaplan-Meier curves of overall survival for the expression(the median expression as a cut-off) of PDHB in **(N)** KICH, **(O)** KIRP, and **(P)** LAML. The Kaplan-Meier curves of overall survival for the expression(the median expression as a cut-off) of SLC31A1 in **(Q)** ACC, **(R)** BLCA, **(S)** BRCA, **(T)** GBMLGG, **(U)** LAML, **(V)** LGG.
(TIF)

**S4 Fig. The forest plot of univariate Cox regression of cuproptosis-related genes for various cancers.**
(TIF)

**S5 Fig. The significant correlations between the cuproptosis-related gene expressions and TMB.** The correlations between TMB and the expression of **(A)** FDX1, **(B)** LIAS, **(C)** LIPT1, **(D)** DLD, **(E)** DLAT, **(F)** PDHA1, **(G)** PDHB, **(H)** SLC31A1. * $p<0.05$, ** $p<0.01$, *** $p<0.001$, **** $p<0.0001$.
(TIF)

**S6 Fig. The significant correlations between the cuproptosis-related gene expressions and MSI.** The correlations between MSI and the expression of **(A)** FDX1, **(B)** LIAS, **(C)** LIPT1, **(D)** DLD, **(E)** DLAT, **(F)** PDHA1, **(G)** PDHB, **(H)** SLC31A1. * $p<0.05$, ** $p<0.01$, *** $p<0.001$, **** $p<0.0001$.
(TIF)

**S7 Fig. The significant correlations between the cuproptosis-related gene expressions and NEO.** The correlations between NEO and the expression of **(A)** FDX1, **(B)** LIAS, **(C)** LIPT1, **(D)** DLD, **(E)** DLAT, **(F)** PDHA1, **(G)** PDHB, **(H)** SLC31A1. * $p < 0.05$, ** $p < 0.01$, *** $p < 0.001$, **** $p < 0.0001$.
(TIF)

**S8 Fig. Cuproptosis-related gene expressions are associated with cells infiltrating in TME of various cancer types.** Correlations between cells infiltrating in TME of various cancer types and **(A)** FDX1, **(B)** LIAS, **(C)** LIPT1, **(D)** DLD, **(E)** DLAT, **(F)** PDHA1, **(G)** PDHB, **(H)** SLC31A1.
(TIF)

**S1 Table. List of abbreviations.**
(DOCX)

**S2 Table. Software details.**
(DOCX)

## Author contributions

**Conceptualization:** Jianpeng Zhou, Chuanlei Wang, Jia Li.

**Data curation:** Jia Li.

**Formal analysis:** Jianpeng Zhou, Chuanlei Wang, Yao Zhi.

**Methodology:** Yao Zhi.

**Project administration:** Jianpeng Zhou, Chuanlei Wang, Jia Li.

**Software:** Yao Zhi.

**Writing – original draft:** Jianpeng Zhou, Chuanlei Wang, Yao Zhi, Jia Li.

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
