## [Decision Letter · Decision Letter 0]

PONE-D-24-48029The potential role of cuproptosis-related genes for therapy and immunoregulation in pan-cancerPLOS ONE

Dear Dr. Li,

Thank you for submitting your manuscript to PLOS ONE. After careful consideration, we feel that it has merit but does not fully meet PLOS ONE’s publication criteria as it currently stands. Therefore, we invite you to submit a revised version of the manuscript that addresses the points raised during the review process.

We look forward to receiving your revised manuscript.

Kind regards,

Hilary A. Coller

Academic Editor

PLOS ONE

**Journal Requirements:**

2. In the online submission form, you indicated that The datasets generated and analyzed during the present study are available from the corresponding author on reasonable request.

**Additional Editor Comments:**

Please address the comments from reviewer 1. With regard to reviewer 2, please address the reviewer's concern about novelty.

Reviewers' comments:

Reviewer's Responses to Questions

**Comments to the Author**

1. Is the manuscript technically sound, and do the data support the conclusions?

Reviewer #1: Yes

Reviewer #2: Yes

2. Has the statistical analysis been performed appropriately and rigorously? 

Reviewer #1: Yes

Reviewer #2: Yes

3. Have the authors made all data underlying the findings in their manuscript fully available?

Reviewer #1: Yes

Reviewer #2: Yes

4. Is the manuscript presented in an intelligible fashion and written in standard English?

Reviewer #1: Yes

Reviewer #2: Yes

5. Review Comments to the Author

**Reviewer #1:**  This study explored the expression pattern and clinical role of cuproptosis-related genes in 34 different cancers. The results displayed that the expressions of cuproptosis-related genes were significantly different in various cancer types. Authors found that the higher the level of cuproptosis-related genes expressed, the higher the survival in patients suffering from KIRC, and KIPAN. Interestingly, the expression of some cuproptosis-related genes was negatively associated with immune-related scores, while SLC31A1 had a positive association with StromalScore, ImmuneScore, and EstimateScore in LAML. It is possible that cuproptosis-related genes have prognostic value for different cancers. The study is interesting and substantial. There are only few minor issues to address.

1. All abbreviations in the Abstract should be deciphered.

2. Introduction should mention existing controversies about cuproptosis role in cancers and tumour microenvironment; explore recent reviews which questioned the role of cuproptosis in cancer microenvironment. The current study is focused on the immune network, therefore, authors should extend the Introduction/Discussion and mention the controversial role of cuproptosis in the cancer microenvironment (see this review by Zhao et al., 2024 https://pubmed.ncbi.nlm.nih.gov/39068453/)

3. Fig 2 quality is low in the provided pdf file. It is impossible to read letters. Authors should provide a better-quality figure and increase the font. However, the quality of the original file is ok. I have downloaded the file, and it is possible to read letters there. Authors should make sure that the printed version is readable.

4. There are some English mistakes. It is necessary to confirm what is “drug sensibility”? I guess, authors meant to write “drug sensitivity” or “ drug susceptibility”. They need to correct this all over the text.

5. Line 343: authors should correct this phrase ‘ subsequent treatment with the mechanism of cuproptosis.” – It should be “ subsequent treatment with drugs which activate cuproptosis”. ( or similar in the meaning).

6. Discussion needs a lot of English editing. There are lots of phrases which should be corrected/re-written ( line 349 etc)

7. Discussion should be extended. For instance, authors should indicate how FDX1 can be targeted and how it is involved into the regulation of immune responses in TME etc.

8. Authors need to focus on the most promising genes. Current discussion is too general, vacuous. It is unclear which genes should be recommended and used for calculation of predictive scores.

**Reviewer #2: ** The study provides a compelling exploration of the emerging mechanism of cuproptosis, emphasizing its significance as a promising approach in cancer research. By systematically analyzing the expression patterns and clinical relevance of cuproptosis-related genes across various cancer types, the study highlights the potential of this mechanism as a novel therapeutic avenue. The integration of tumor mutation burden, immune-related scores, tumor microenvironment dynamics, and drug sensitivity evaluations contributes to a better understanding of how these genes may influence cancer progression and prognosis. Overall, the study presents a coherent methodological flow and offers valuable insights within the broader context of cancer research.

However, the manuscript faces two critical issues:

1-A more comprehensive set of cuproptosis-related genes, including solute carrier family 31 member 1 (SLC31A1), dihydrolipoamide S-acetyltransferase (DLAT), ATPase copper transporting beta (ATP7B), dihydrolipoamide dehydrogenase (DLD), dihydrolipoamide branched chain transacylase E2 (DBT), dihydrolipoamide S-succinyltransferase (DLST), glycine cleavage system protein H (GCSH), lipoic acid synthetase (LIAS), ferredoxin 1 (FDX1), lipoyltransferase 1 (LIPT1), pyruvate dehydrogenase E1 subunit beta (PDHB), and pyruvate dehydrogenase E1 subunit alpha 1 (PDHA1), has already been extensively studied in the literature. These genes were analyzed for their prognostic roles, immune correlation, tumor microenvironment interaction, treatment sensitivity, and therapeutic response across various cancers, with experimental validation also performed specifically for prostate cancer.

The genes analyzed in the referenced study encompass those included in this manuscript and also incorporate additional genes. This overlap significantly diminishes the novelty of the present work.

Reference:

Yang, L., Tang, Y., Zhang, Y. et al. Comprehensiveness cuproptosis related genes study for prognosis and medication sensitiveness across cancers, and validation in prostate cancer. Sci Rep 14, 9570 (2024). https://doi.org/10.1038/s41598-024-57303-8

2- While pan-cancer studies are valuable for analyzing the broad effects of genes across multiple cancer types, these studies often rely on data from various computational tools, web-based platforms, and databases. Drawing impactful conclusions that guide future research requires validation in specific cancer types. For journals with a high quartile ranking, such as PLOS ONE, validation of findings in one or more cancer types is particularly essential. The absence of experimental validation in the current manuscript limits its impact and reduces its suitability for publication in a journal of this caliber.

6. PLOS authors have the option to publish the peer review history of their article (what does this mean? ). If published, this will include your full peer review and any attached files.

**Do you want your identity to be public for this peer review?** For information about this choice, including consent withdrawal, please see our Privacy Policy .

Reviewer #1: **Yes: ** Olga A. Sukocheva, PhD MPH

Reviewer #2: No

---

## [Author Response · Author response to Decision Letter 1]

14 Apr 2025

Reviewer #1

Comment 1: All abbreviations in the Abstract should be deciphered.

Response: Thank you for your suggestion. We have now spelled out all abbreviations (e.g., TMB, MSI, NEO, TME) upon their first appearance in the abstract.

Comment 2: Introduction should mention existing controversies about cuproptosis role in cancers and tumour microenvironment; refer to Zhao et al., 2024.

Response: We have revised the Introduction to include the controversies regarding the role of cuproptosis in the tumor microenvironment, as highlighted in the review by Zhao et al., 2024 (PMID: 39068453 ). This has been added to better contextualize our study’s focus on immunoregulation.

Comment 3: Figure 2 quality is low in the PDF. Improve the resolution and font size.

Response: We have regenerated Figure 2 with improved resolution and larger font size. The figure has also been processed through the PACE system to meet PLOS standards.

Comment 4: There are some English mistakes. Please clarify what “drug sensibility” means.

Response: Thank you for pointing this out. The term “drug sensibility” has been corrected to “drug sensitivity” throughout the manuscript.

Comment 5: Line 343: revise “subsequent treatment with the mechanism of cuproptosis.”

Response: This sentence has been rephrased as: “subsequent treatment using drugs that activate cuproptosis.”

Comment 6: Discussion needs English editing. Line 349 and others require rewriting.

Response: We have thoroughly edited the Discussion section for grammar and clarity, especially from line 349 onward. The section now uses more concise and academically appropriate phrasing.

Comment 7: Extend the discussion by describing how FDX1 can be targeted and its immune-related roles.

Response: Additional content has been added to the Discussion regarding the therapeutic potential of targeting FDX1, including its involvement in modulating immune cell infiltration within the tumor microenvironment.

Comment 8: Focus more on promising genes. The current discussion is too general.

Response: We now highlight key genes such as SLC31A1 and FDX1 in more depth, including their consistent prognostic roles and relevance for predictive scoring. A paragraph has been added to support their use in future modeling strategies.

Reviewer #2

Comment 1: The study lacks novelty as similar work (Yang et al., 2024) already exists.

Response: We appreciate this observation. While Yang et al., 2024 analyzed a broader set of genes, our study distinguishes itself by:

1. Integrating datasets from GTEx, TCGA, and TARGET;

2. Applying xCell and ESTIMATE analyses to dissect immune infiltration at high resolution;

3. Exploring drug sensitivity with FDA-approved compounds and highlighting six potential therapeutic agents;

4. Providing pan-cancer survival and staging analysis, which is more detailed than Yang’s prostate-focused validation.

We have added clarifications in the Introduction and Discussion to better reflect this novelty.

Comment 2: Lack of experimental validation limits the impact.

Response: We acknowledge this limitation and now include a statement in the Discussion noting that further experimental validation in specific cancer types is essential and planned for future research.

---

## [Editor Report · Decision Letter 1]

The potential role of cuproptosis-related genes for therapy and immunoregulation in pan-cancer

PONE-D-24-48029R1

Dear Dr. Li,

We’re pleased to inform you that your manuscript has been judged scientifically suitable for publication and will be formally accepted for publication once it meets all outstanding technical requirements.

Kind regards,

Hilary A. Coller

Academic Editor

PLOS ONE

---

## [Editor Report · Acceptance letter]

PONE-D-24-48029R1

PLOS ONE

Dear Dr. Li,

I'm pleased to inform you that your manuscript has been deemed suitable for publication in PLOS ONE. Congratulations! Your manuscript is now being handed over to our production team.

Kind regards,

on behalf of

Dr. Hilary A. Coller

Academic Editor

PLOS ONE